# Artificial Intelligence Assisted Heating Ventilation and Air Conditioning Control and the Unmet Demand for Sensors: Part 2. Prior Information Notice (PIN) Sensor Design and Simulation Results

**DOI:** 10.3390/s19153440

**Published:** 2019-08-06

**Authors:** Chin-Chi Cheng, Dasheng Lee

**Affiliations:** Department of Energy and Refrigerating Air-Conditioning Engineering, National Taipei University of Technology, Taipei 10608, Taiwan

**Keywords:** artificial intelligence (AI) assisted HVAC control, prior information notice (PIN) sensor, prediction accuracy, conditional probability, energy savings

## Abstract

The study continues the theoretical derivation from Part 1, and the experiment is carried out at a bus station equipped with six water-cooled chillers. Between 2012 and 2017, historical data collected from temperature and humidity sensors, as well as the energy consumption data, were used to build artificial intelligence (AI) assisted heating ventilation and air conditioning (HVAC) control models. The AI control system, in conjunction with a specifically designed prior information notice (PIN) sensor, was used to improve the prediction accuracy. This data collected between 2012 and 2016 was used for AI training and PIN sensor testing. During the hottest week of 2017 in Taiwan, the PIN sensor was used to conduct temperature and humidity data predictions. A model-based predictive control was developed to obtain air conditioning energy consumption data. The comparative results between the predictive and actual data showed that the temperature and humidity prediction accuracies were between 95.5 and 96.6%, respectively. Additionally, energy savings amounting to 39.8% were achieved compared to the theoretical estimates of 44.6%, a difference of less than 5%. These results show that the experimental model supports the theoretical estimations. In the future, a PIN sensor will be installed in a chiller to further verify the energy savings of the AI assisted HVAC control.

## 1. Introduction

The heating, ventilation and air conditioning (HVAC) system is able to provide a satisfied thermal comfort and acceptable indoor air quality for occupants by adjusting and changing the air condition of occupied buildings. To reach this goal, the required processes include heating, cooling and ventilation, as well as humidification and dehumidification processes. These processes can be accomplished through components that include a compressor, fan, condenser, evaporator, and expansion valve. For the proper operation of these processes and components, the suitable sensors to detect the status of the environment, coolant and components are quite necessary. The general sensors employed by HVAC systems include temperature, humidity, pressure, wind velocity anemometers, as well as a flow meter. To detect the thermal comfort or occupancy behavior, ultrasonic sensors, CO_2_ detectors, wearable devices, smart phones, radio frequency identification devices (RFIDs), passive infra-red (PIR) sensors, and wireless sensor networks are also utilized. The progress of these sensors before 2016 has been surveyed in our previous paper [1]. Between 2016 to 2019, oxygen sensors [2], internet of things (IoT) frameworks with conventional sensors [3,4], particle counters, CO, NO_2_ sensors [5], smart sensors incorporating a microprocessor chip [6], visual sensors [7], thermal flow sensors [8] and infra-red (IR) cameras [9] have been utilized to detect the thermal comfort, occupancy preference, and to eliminate measurement errors and reduce energy consumption. The detection of information by developed sensors will be fed back to the HVAC system for compensating the error between the set values and the measured ones. This is the so-called difference-causing control system, which is widely used in the On-Off and proportional–integral–derivative (PID) control algorithms of HVAC systems. The difference-causing control feedback systems will result in energy waste and human uncomfortableness [10]. In this research, the authors present a novel notation of the predictive sensor and feedforward control by utilizing artificial intelligence (AI) tools, able to accurately predict the demand of air conditioning space and to adjust the output of the air conditioner in advance.

Artificial intelligence (AI) had been present since 1956. AI technology presents machine intelligence and acts like human behavior through mathematical coding and mechanical works. Through the developing progress of AI technology, there are 18 developed AI technologies applied on the HVAC control systems [11], including neuro networks (NN), fuzzy, the model-based predictive control (MPC), distributed AI, the multi-agent system (MAS), the genetic algorithm (GA), support vector machines (SVM), R, model-based controls, deep learning (DL), the knowledge-based system (KBS), case-based reasoning (CBR), particle swarm optimization (PSO), the artificial fish swarm algorithm (AFSA), the hidden Markov model (HMM), the radial basis function (RBF), the k-nearest neighbor (KNN), and autoregressive exogenous (ARX). Among them, the most well-known AI tools are neuro networks (NNs), including artificial neuro networks (ANNs), recurrent neuro networks (RNNs), spiking neuro networks (SNNs), and wavelet ANNs. These AI tools function either with one single algorithm, or combine with other algorithms (two or more) to provide the designed solutions for HVAC systems. However, only two methodologies, which are the forecasting-and-optimization and the predictive controls, have become mainstream elements of HVAC controls over the past twenty years. Hence, the AI-assisted HVAC control is a method of using predication to control and improve overall performance.

However, based on the presented data in Part 1, the energy savings of AI-assisted HVAC control systems are less than for those equipped with traditional energy management system (EMS) controlling techniques. This is because most of the prediction uncertainties of AI tools are larger than those of most of the existing sensors. Furthermore, they cannot meet the required demand for AI functionality. A literary review related to the prediction uncertainty of controlling AI-assisted HVAC systems was conducted, including energy consumption, the heating or cooling load, human behavior, the thermal comfort index and the weather forecast. The test sites include households [12,13,14,15,16,17,18], grid systems [19,20,21,22], and some combinations with building energy management systems (BEMSs) [23,24]. The estimated periods of the prediction function include short/long-term, such as hourly [13,16,17,25,26,27,28], daily [29] and long-term [27]. The most utilized AI functions are NNs and related tools [14,16,17,18,21,23,25,26,27,29,30,31,32,33,34,35,36,37]. The second useful AI functions are fuzzy [34,38,39,40,41,42], SVM [30,36,43,44] and GA [34,45,46]. Referring to the literatures above and from Part 1, the prediction uncertainty for the energy consumption, heating or cooling load is 7.46%, the prediction uncertainty for human behavior and thermal comfort is 14.50%, and the weather forecast uncertainty is 3.50%. The prediction uncertainties for the temperature and humidity sensors, which are widely used in HVAC systems, are between 2.77 to 4.0%. Besides the weather forecast results, the prediction uncertainty of all the other criteria were higher than those of typical HVAC control sensors, clearly showing that the data gathered from the sensor is more accurate than the predicted data.

Therefore, the capability to control the HVAC system performance through AI tools has been limited. This is due to the degree of prediction uncertainty being higher than that of a sensor. If there is a way to effectively improve the prediction accuracies, the overall HVAC control performance could be improved. In this research, Bayes’ theorem is utilized in order to improve the predictive accuracy. The theorem describes that the conditional probability of the random Events A and B could be calculated based on the associations between Events A and B and their time sequences. The probability of Event A is predicted accurately through a step-by-step convergence calculation under the condition of standard likelihood. Referring to Bayes’ theorem and applying conditional probability, an innovative priori information notice (PIN) sensor is provided to uplift the prediction accuracy and the energy savings of an air conditioning plant. After elevating the prediction accuracy of AI tools through the assistance of the PIN sensor, the AI-assisted HVAC control is able to reach energy savings of 57% and 44.64%, compared to the On-Off and PID control, respectively. Based on the above theoretically estimated results, an experimental case was selected, which involved installing an innovative PIN sensor and calculating the energy savings through a simulation. The results were then compared with the theoretical estimation results.

In short, this study provides the notation of a predictive sensor and feedforward control algorithm for an HVAC system. By utilizing Bayes’ theorem and applying the conditional probability, a novel PIN sensor is able to uplift the prediction accuracy and energy savings of an air conditioning plant. The accumulated data of five years in the selected test site, starting in 2012, is utilized to train the PIN sensor and calculate the energy savings. Compared with the conventional On-Off and PID control, the presented AI-assisted HVAC control is able to reach an accurate prediction and better energy savings.

## 2. The Design and Verification Procedure

In this study, Bayes’ theorem and conditional probability are utilized to design an innovative PIN sensor and uplift the prediction accuracy and energy savings of an air conditioning plant. The designing procedure includes the following five steps:

### 2.1. Big Data Collection

In this study, a four-story, 25,996 m^2^ facility equipped with water-cooled chillers was utilized to conduct experiments and collect data, serving as a means for the development basis of an innovative PIN sensor. The collected data included the electricity usage of facility, the electricity usage of the 6 water-cooled chillers, as well as the overall temperatures, humidity levels, in addition to climate data gathered from an outside weather station. The data was collected for five years between 2012 and 2017, and that of the first four years was used for the purpose of AI training. The data collected in the last year was used to verify the prediction accuracies and to calculate the energy savings.

### 2.2. Data Feature Analysis

As mentioned previously, the collected data in the first four years was used for AI training to discover data features. In this study, an innovative AI tool was utilized to recognize graphs. Then, an AI tool provided by the Microsoft Azure platform was utilized for the predictive control after training by using the first four years of data.

### 2.3. Priori Information Notice Sensor Design and Application

In order to further improve on the aforementioned AI prediction accuracy, the study proposes an innovative circuit design, known as the priori information notice (PIN) sensor, which combines with an existed sensor to have the ability to make predictions based on priori information. Based on Bayes’ Theorem, the prediction accuracy of the PIN sensor could be greatly improved under the condition of knowing a priori event. Therefore, the combination of the PIN sensor and AI training tool is expected to uplift the prediction accuracy in the same way as the sensing accuracy of the existing sensor. The performance of the predictive control is also improved.

### 2.4. Model Base Control and AI Implementation

The predictive control methodology for the HVAC equipment is determined by a knowledge base. Through the accumulation of 5 years of air conditioning operational data, a plant model is established in order to conduct the model base AI-assisted HVAC control. An element primary to the AI implementation is to use the aforementioned PIN sensor in a predictive pattern to control the entire plant model, replacing the original method of feedback control with the expectation of achieving a better control performance.

### 2.5. Control Performance Verification

We obtained the exact energy savings of the air conditioning system by using the PIN sensor and predictive control under the operational conditions of the hottest month of summer of 2017. The simulation was designed to obtain real energy savings and verify the hypothesis discussed in Part 1, which declares that using a PIN sensor could solve the issue of unmet sensor demands and increase energy efficiency. In addition, the air conditioning system settings at the test site were used to confirm the air conditioning system’s On-Off or PID control, and after making further changes to the AI assisted control, they were used to calculate specific energy savings.

## 3. Test Site—An Air Conditioning Plant with Six Water-Cooled Chillers

The simulated study was conducted in a real complex building located at the Taipei Bus Station, a major transfer hub located in Taipei City, Taiwan, as shown in Figure 1a. A total of 36 operators approved by the Ministry of Highways and Transportation operates from this hub, servicing 170 routes, 99 of which start from Taipei. There are another 15 operators servicing 65 routes around Taipei Bus Station. On a daily average, there are 2289 departures from the station and 2427 departures on peak days. The average number of daily commuters is approximately 80,000.

The transfer station itself is a four-story, 25,996 square meter facility. The ticket hall is located on the first floor. There are 16 platforms on each floor, and a totality of 48 platforms from the second to fourth floors. Air conditioning for the entire building is provided by 6 water-cooled chillers, as shown in Figure 1b. Among them, chillers numbered 1–2 have a freezing capacity of 100 refrigeration tons (RTs), and chillers numbered 3–6 have a freezing capacity of 80 RTs. Each chiller was equipped with a digital meter that records the energy usage data. In addition, a digital meter was installed in the main controller board to monitor the entire building’s energy usage.

Besides the monitoring energy usage data, located throughout the transfer station’s four floors’ waiting area space, there were 16 networked temperature and humidity sensor modules that provide the temperature and humidity data for each floor, as shown in Figure 1c,d. Along with these sensor modules, a carbon-dioxide sensor was placed on each floor of the facility to evaluate the indoor air quality. Finally, a weather station located on the roof of the building was used to provide the outside temperature, humidity and wind speed data.

The study used the accumulated data of five years, starting in 2012, regarding the mentioned energy usage and indoor/outdoor temperature and humidity data. The study will reference the 5 steps discussed (AE) to verify if the AI-assisted HVAC control can achieve energy savings in a real air conditioning case, specifically finding out whether or not the PIN sensor can effectively predict the indoor and outdoor weather conditions that are necessary for the chiller control in order to achieve energy savings.

At the case study location, the inverter control was applied during the chiller water delivery. In addition, the cooling capacity was regulated by the PID control. Hence, this study focuses on the PID controlled chiller plant, and its historical data (as the energy consumption baseline) is used to calculate the energy savings. In addition, whether or not the energy savings could reach the theoretical estimation of 44.64% is also interesting.

## 4. PIN Sensor Design

The working principles of the PIN sensor depend on the conditional probability. The PIN sensor can make predictions based on analyzing the incoming signal and continuously modifying the signal. Continuous modifications are made by looking at correlations between two events designated as Event A and Event B. The initial prediction is called the prior probability. When there are occurrences of Event A, the initial prediction (prior probability) will then be continuously modified according to Event B to determine what is called the posterior probability.

Based on Bayes’ theorem [3], the posterior probability will become higher since Events A and B have a strong correlation with each other. Therefore, using the prior and posterior probabilities can yield accurate predictions of future signals. With respect to the control problem discussed in this study, the s parameter is designated to be the highly correlated events, and is defined as [47]:(1)s=σ+jω
where σ and ω are the real and imaginary parts of the s parameter, respectively. Laplace transform is able to convert the time-domain function into the s-domain function. In this study, the s parameter converts the sensor output signal S(t) into the needed Events A and B for an accurate prediction. This can be described in the following formula:(2)$$S\left(t\right)=\exp\left(\mathrm{st}\right)=e^{\sigma t}\cdot e^{j\omega t}$$
where S(t) denotes the sensor signal output. Events A and B can be identified as the trend and frequency of the sensor signal output, as shown in Formula (3):(3)$$\begin{array}{c} e^{\sigma t}\to \mathrm{Event}\;A:\;\mathrm{time}\;\mathrm{trend}\;\mathrm{in}\;\mathrm{proportional}\;\mathrm{to}\;\sigma \\ e^{j\omega t}=\;\cos\left(\omega t\right)+j\cdot \sin\left(\omega t\right)\to \mathrm{Event}\;B:\;\mathrm{frequency}\;\mathrm{output} \end{array}$$

The designed PIN sensor is able to detect and convert the sensor output signal into the time trend and frequency, which serve as Events A and B. The conditional probability of Event A_j_ under the condition of the occurrence of Event B is described as follows:(4)$$P\left(A_{j}|B\right)=\frac{P\left(A_{j}|B\right)\cdot P\left(A_{j}\right)}{P\left(A_{j}|B\right)\cdot P\left(A_{j}\right)+P\left(A_{j}|\neg B\right)\cdot \left(1-P\left(A_{j}\right)\right)}$$
where A_j_, j = 1, …k is a collection of time trends, and ¬B is the complementary of Event B. $P\left(A_{j}\right)$ is the prior probability. For any other occurred Event B, *P*(*B*) > 0, the posterior probability of A_j_ under the occurrence of Event B can be calculated by Equation (4). The posterior probability can be used to increase the prediction accuracy. To achieve this, the PIN sensor continuously converts sensor output signals into time trends and frequencies as the posterior Events A and B, and makes continued revisions of $P\left(A_{j}|B\right)$ and $P\left(A_{j}|\neg B\right)$. This will gradually increase the prediction accuracy, and when used in tandem with AI it is able to advance the predictive control. The following diagram illustrates the architecture of the PIN sensor design.

Figure 2 shows the PIN sensor design, which is a novel design to upgrade the current sensor and frequency acquiring circuit as the predictor for the future predictive control. The analog/digital converter, as shown at the top of Figure 2, converts the analog signal into a digital one for the record and display. The PIN sensor, as shown at the bottom of Figure 2, converts the sensor output signal S(t) into the two events of trend and frequency. The trend filter, as shown at the bottom left of Figure 2, is composed of a diode and a variable capacitor. The Event A is composed of the trend output of the sensor output signal, which is the applied voltage value of the variable capacitor when the current passing through the diode is zero. The value of the variable capacitor is adjusted by a feedforward control. The frequency acquiring circuit, as shown at the bottom right of Figure 2, is composed of an OP amplifier, RLC circuit and FFT analyzer. The Event B is composed of the frequency output of the sensor output signal, which is amplified by the OP amplifier, and acquired by the RLC circuit and FFT analyzer. The further application of the PIN sensor in Figure 2 for predicting the tendency of the trend and frequency will be illustrated in the following section.

The trend and frequency outputs of the sensor output signal, analyzed by the PIN sensor, are two highly correlated events. Based on Bayes’ theorem, the prediction accuracy of the PIN sensor is improved by calculating the conditional probability of the two Events A and B. Furthermore, used in conjunction with AI, a higher predictive control performance can be realized.

## 5. Simulation Model and Applied Tools

The PIN sensor design utilizes the Matlab Simulink tool [48]. The value of a variable capacitor and the magnification of an OP amplifier are calculated from the collected data of the test site during the period of 2012~2016, including the energy consumption of six water-cooled chillers, values of 16 internal temperature and humidity sensors, as well as the temperature and humidity sensor from the exterior weather station.

After the completion of the PIN sensor design, the collected data gathered in 2017 was used to conduct a blind test. Specifically, 3 items of data regarding P(Aj)$,\;P\left(A_{j}|B\right)$ and $P\left(A_{j}|\neg B\right)$ were examined to test the predictive accuracy. At 1:18 pm, on September 27, 2017, a high temperature of 38.6 °C was recorded, which happened to be the highest temperature recorded in 121 years of weather observation in Taiwan. These extreme conditions undoubtedly had an effect on the prediction accuracy as well as the predictive control performance. It was during the hottest week of 2017, between September 25th and October 1^st^, that the data for the study was collected to conduct the simulation and energy saving performance verification. Under these harsh conditions, the designed PIN sensor was verified while simultaneously using real data to calculate the energy consumption yielded by the AI-assisted HVAC control.

The study used the Microsoft Azure platform as the AI tool, along with the PIN sensor, to conduct the predictive control. The mainly utilized AI tools were Azure Machine Learning kits, carried out by Python SDK, along with Visual Studio Core. After using the collected data to carry out machine learning, the temperature and humidity data during September 2017 were predicted. After comparing with the measured data if the temperature and humidity differences were under 0.1 °C and 0.5%, respectively, the predicted data could be considered as being an accurate prediction.

The predicted temperature and humidity data can be applied directly to a model to carry out the predictive control. Regarding our test case, the COMSOL Multiphysics modeling software was first used to simulate the entire building and calculate the thermal resistance (R_th_) between 16 sensors and the outside environment. Then, on the basis of the daily traffic data provided by the management office of the case study, the real-time pedestrian movement MC(t) was estimated. Through the temperature and humidity data from 16 sensors (T1~T16 and RH1~RH16) and the weather station (Tamb and Hamb), the removed heat q’(t) and an HVAC control model could be calculated. The building simulation and model for the AI control is shown in the illustration below.

The test site simulation shown in Figure 3 can be expressed by the following formula:(5)$$\begin{array}{cc} q^{\prime }\left(t\right)= & \left[\begin{array}{ccc} \frac{\mathrm{MC}{\left(t\right)}_{1F}}{4} & \cdots & \frac{\mathrm{MC}{\left(t\right)}_{4F}}{4} \end{array}\right]\left[\begin{array}{c} \frac{\mathrm{dT}1\left(t\right)}{\mathrm{dt}}\\ \begin{array}{c} \vdots \\ \frac{\mathrm{dT}16\left(t\right)}{\mathrm{dt}} \end{array} \end{array}\right]+\left[\begin{array}{ccc} \frac{1}{\mathrm{Rth}1} & \cdots & \frac{1}{\mathrm{Rth}16} \end{array}\right]\left[\begin{array}{c} T1\left(t\right)-\mathrm{Tamb}\left(t\right)\\ \begin{array}{c} \vdots \\ T16\left(t\right)-\mathrm{Tamb}\left(t\right) \end{array} \end{array}\right]\\ & +{m^{\prime }}_{\mathrm{air}}\cdot [(h\left(T_{\mathrm{amb}},{\;\mathrm{RH}}_{\mathrm{amb}}\right)\;-h\left(\mathrm{Average}\left[\begin{array}{c} T1\\ \vdots \\ T16 \end{array}\right],\mathrm{Average}\left[\begin{array}{c} \mathrm{RH}1\\ \vdots \\ \mathrm{RH}6 \end{array}\right]\right)] \end{array}$$

The first part of Equation (5) describes the real-time thermal mass change caused by pedestrian movement, while the second and third parts describes the heat transmitting through the thermal resistance (wall) and the air movement, respectively, from the outside temperature and humidity. Based on Equation (5), the temperature T1~T16 and relative humidity RH1~RH16 predicted by AI are able to calculate the heat flux needed for the air conditioning system. That enables a model based predictive control, and the energy consumption E_pred_ can be calculated by:(6)$$E_{\mathrm{Pred}}\left(t\right)=3.5\cdot q^{\prime }\left(t\right)/\mathrm{Average}\left({\mathrm{COP}}_{\mathrm{Chiller}1},\ldots ,{\mathrm{COP}}_{\mathrm{Chiller}6}\right)$$
where COP, with respect to every chiller, is directly measured online, and the coefficient 3.5 is used to convert the unit RT into a kW value [49,50]. During the simulation, the COP also varies with the outside temperature and humidity.

## 6. Results and Discussions

Following the five research steps, as illustrated in the introduction section, the PIN sensor design and performance of the AI-assisted HVAC control are presented.

### 6.1. Big Data Collection

Between 2012 and 2017, a total of 630,072,000 data was collected to design the PIN sensor and verify the AI-assisted HVAC control performance. Figure 4a shows the temperature sensor data at the 1F lobby of the test site location on one day of 2012.

The basis of the PIN sensor design is that the daily collected sensor output data can be efficiently converted into the trend and frequency, which are the DC and AC components of the sensor output data. According to the variation of trend, the collected sensor output data can be divided into 6 segments, as shown in Figure 4b–h. After further analysis of each segment, the trend and frequency response of the sensor output data have a close relationship, which is the design basis of the PIN sensor. Matlab Simulink is utilized to design the analog and digital circuitry, and to verify whether or not the designed circuit can convert the sensor output signal into the segment length time, trend and frequency.

### 6.2. Data Feature Analysis

The AI insight can be achieved by the dig data feature through analyzing the 504,057,600 records of temperature and humidity data within 4 years, as shown in Figure 5. This is the foundation for the AI-assisted HVAC control.

In Figure 5, a 6-dimensional diagram is able to express big data in terms of the trend, with respect to the coefficients a and b, as well as the frequency, with respect to the coefficients of amplitude and phase degree. Figure 5a shows that a total of 99.7% of the 504,057,600 data uses four standard types to predict the trend, which include a∗exp(−bt), a∗exp(+bt), +b∗ln(t)+a and −b∗ln(t)+a. Figure 5b shows that the data trend at each time t has a corresponding frequency spectrum. Based on the link between the trend and frequency, and the modification of the prediction following Bayes’ theorem, an accurate prediction can be achieved.

### 6.3. PIN Sensor Design and Application

The 504,057,600 instances of big data that were accumulated between 2012 and 2016 were used to conduct the AI training for the PIN sensor, which was then used to predict the data for the year of 2017. The predicted values, including the temperature and relative humidity of 2017, were compared with the real data obtained from the real site records. Regarding the hottest week recorded in 2017, the designed PIN sensor was used to predict the air conditioning electricity consumption of the case study. First, the accumulated big data of the past 4 years was used to go through machine learning, and AI outputted the prediction values. The prediction accuracy rate is defined as the rate of the predicted data located within a 5% range of the accurate value. Second, the process utilized conditional probability, which improved the PIN sensor prediction accuracy to 95.5%. The same method was applied to the test site’s 16 temperature and humidity sensors to conduct the prediction. First, through machine learning only, the outputted AI prediction accuracy rate was 85%. After utilizing conditional probability, the PIN sensor prediction accuracy rate reached 96.6%.

In summary, the PIN sensor design can increase the prediction accuracy rate, and the uncertainty level can be as low as 3.4~4.5%. This is already in line with the sensing accuracy of the temperature and humidity sensors, which was between 2.77~4%. Thus, we could verify whether or not the energy efficiency of the AI-assisted HVAC control was better than that of the PID control or the On-Off control, which was 44.64~57%.

### 6.4. Model Base Control and AI Implementation

This step aims to design a control system integrated within the PIN sensor and AI control. Figure 6 presents the working principles of the AI-assisted HVAC control system.

The data feature in Figure 5 shows that every temperature and humidity data of the sensing points corresponds to the related trend and frequency spectrum. The working principles of the AI-assisted HVAC control are shown in Figure 6a–f, using the frequency spectrum similarity to make amendments to the trend. The initial trend guess was used for the priori probability; then, by using the frequency spectrum outputted by the PIN sensor, the modified trend is obtained for the posterior probability in order to establish the conditional probability and to improve the prediction accuracy. This is used to more precisely predict the following temperature and humidity values. The results are then utilized to accomplish the model-based AI assisted HVAC control according to Equation (5).

### 6.5. Control Performance Verification

The study set out to validate the energy savings of the designed PIN sensor and AI-assisted HVAC control during the hottest week of 2017. First, the coefficient of performance (COP) of each chiller located within the case study facility was measured, and by using Equation (6) the energy consumption of the AI-assisted HVAC control system was calculated. Then, the real energy consumption of each chiller operated by the PID control system was measured by a digital meter. The energy savings of the AI-assisted HVAC control system will be verified by comparing these two values. The results are shown in Figure 7.

As shown in Figure 7, the water-cooled chillers located in the case study facility use invertor motors and PID control algorithms, and, in theory, after these PID control algorithms were replaced by the AI-assisted HVAC control, 44.64% savings on the energy consumption should be possible. Figure 7 shows the actual conditions of the PID control that occurred on the Saturday during the hottest week of 2017. The peak traffic of people coming and going within the case study facility caused the PID control oscillation, as shown by the darker lines of Figure 7 from the 7200 to 8640 min mark. When the AI-assisted control was used, because of its ability to effectively predict the load on the air conditioning system in the next time point and because of its more stable output compared to the PID control, the total energy savings throughout the day reached 39.8%. Compared to the initial estimated value of 44.64%, there was only a difference of less than 5%. The statistics for the entire week show that the energy savings for the AI-assisted HVAC control was 34.66%, and this result, compared to the theoretical predictions, was over 10%. According to Part 1 of the theoretical analysis, this is the highest proportion of energy savings when the different value of the control performance index NHI is at its maximum value. Because of this, during the hottest conditions, using the PID control will generate a control oscillation. However, using the AI-assisted control is relatively more stable, and the energy savings do approach the theoretical values of 44.64%. At other times, using the PID control still maintains a certain control efficiency and energy savings. In this case, the decrease of energy savings caused by the AI-assisted control can still be regarded as being within the scope of the theoretical speculations.

Through these five experimental steps (A–E) described in the introduction, the study was able to verify that on the Saturday of the hottest week ever recorded during peak traffic times, the AI-assisted control had energy savings of 39.8%, when compared to the PID control. This value was on a par with the previously made theoretical estimations. In the future, with the support from the manager of the case study facility, an actually produced innovative PIN sensor will be installed into a water-cooled chiller in Part 3 of this study. Additionally, the study will conduct the actual control of a water-cooled chiller to further conduct an integrated analysis of the theory, the simulation work, and the actual control of the device.

## 7. Conclusions

The study proposed an innovative sensor design that, in conjunction with the AI-assisted control algorithms, is to be used in a 4-story case study facility at the Taipei Bus Station equipped with 6 water-cooled chillers for conducting control simulations. Following the experimental steps described in steps A to E, the AI-assisted control (comparable to the existing PID control) showed energy savings of 39.8%. The key is the designed PIN sensor, which is able to convert the sensor output signal into two events: the trend and frequency. Because of the correlations between the two events of trend and frequency, and according to Bayes’ theorem, the output signals of the next time point could be accurately predicted. Regarding the temperature and humidity values, after going through a series of blind tests, their values had an accuracy rate of 95.5% and 96.6%, respectively. Being able to make accurate predictions is the basis for reaching theoretical energy savings. In the future, a prototype PIN sensor and corresponding AI controller will be applied for controlling a water-cooled chiller, combining theory with practice in order to verify the performance of the PIN sensor and AI-assisted HVAC control.

## Figures and Tables

**Figure 1 sensors-19-03440-f001:**
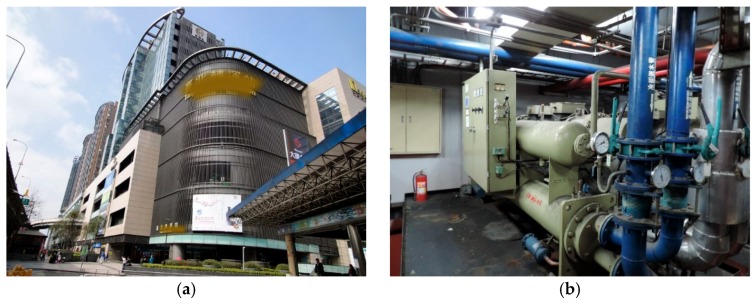
A real test site for presenting artificial intelligence (AI)-assisted heating ventilation and air conditioning (HVAC) control performances. (**a**) The case study facility; (**b**) the chiller system including 6 chillers installed in the building; (**c**) a power meter for measuring the energy consumption; (**d**) a networked temperature/humidity sensor module for detecting the necessary parameters for the HVAC control.

**Figure 2 sensors-19-03440-f002:**
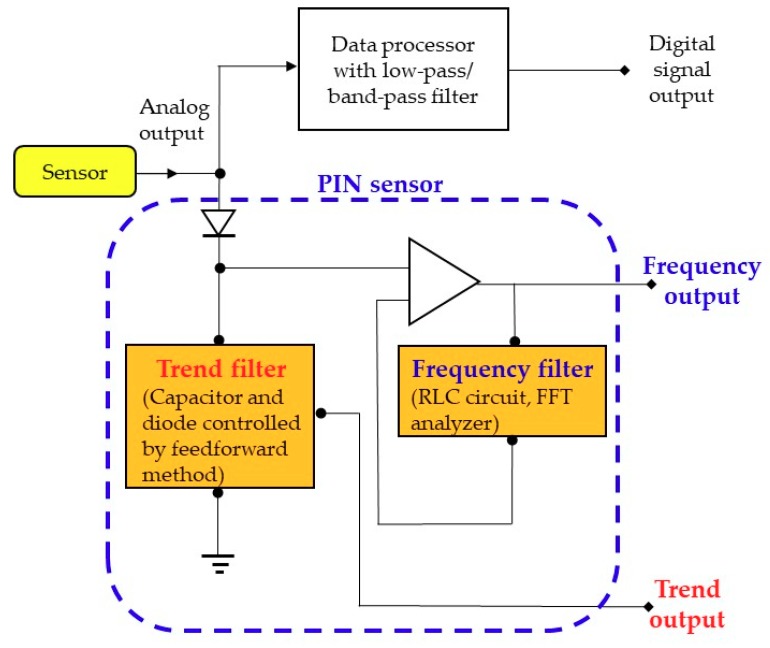
The PIN sensor design for increasing the prediction accuracy based on the conditional probability and Bayes’ theorem.

**Figure 3 sensors-19-03440-f003:**
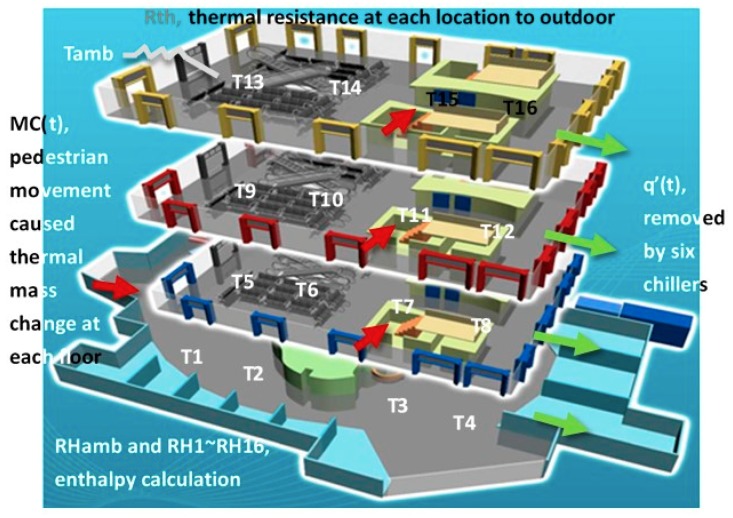
The test site simulation and related parameters for building a control model.

**Figure 4 sensors-19-03440-f004:**
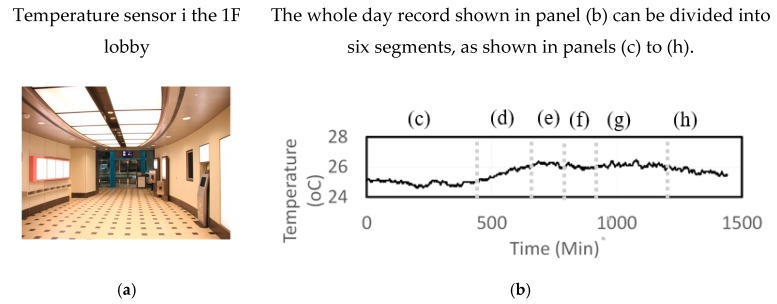
Big data collection for the PIN sensor design and AI control performance test: (**a**) a sample of one temperature and humidity sensor installed in the 1F lobby; (**b**) The real time data collection of a day; (**c**–**h**) The one-day data can be divided into six segments, and each segment data has different trends and frequencies.

**Figure 5 sensors-19-03440-f005:**
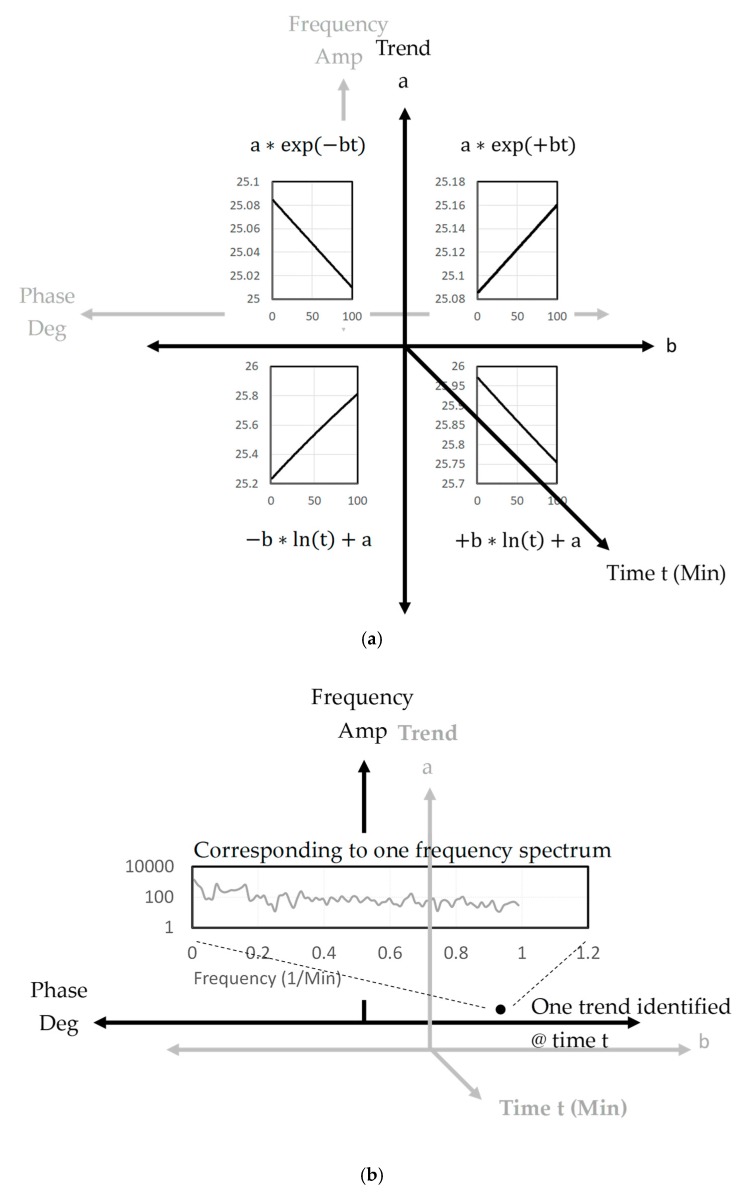
Two features of big data, including (**a**) the signal trend, can be expressed by four standard types; (**b**) one frequency spectrum corresponding to one signal trend.

**Figure 6 sensors-19-03440-f006:**
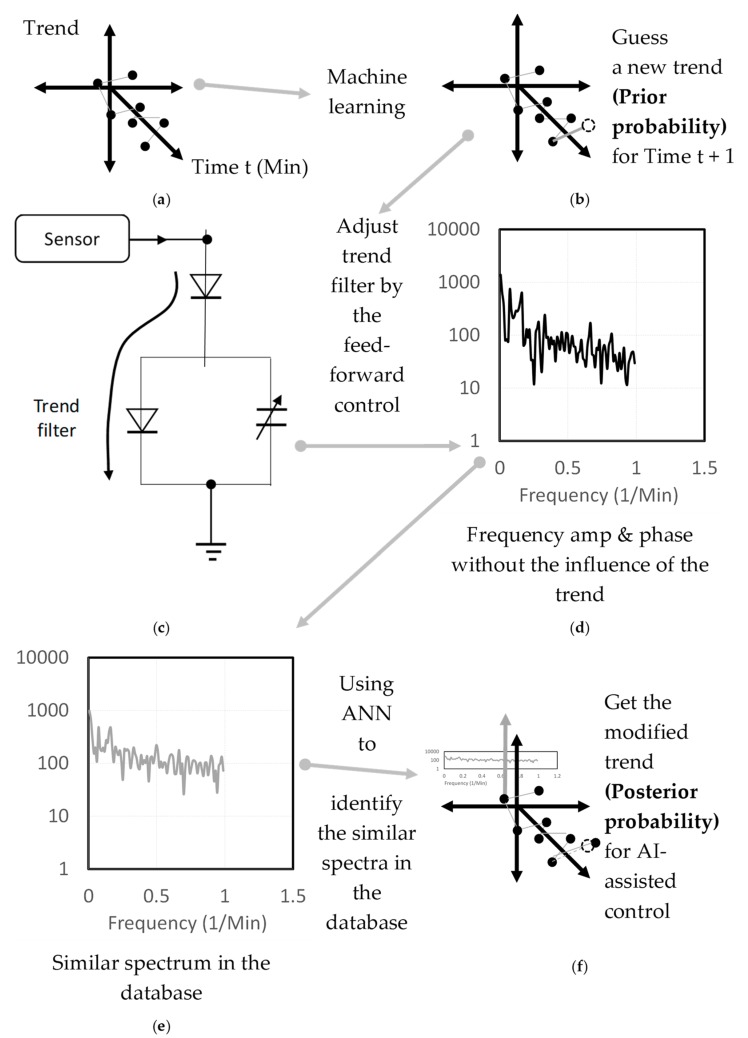
Working principles of the AI-assisted HVAC control (**a**) Trend from the big data collection; (**b**) Initial guess of the trend by machine learning; (**c**) Adjustment of the trend filter for the PIN sensor; (**d**) Frequency output; (**e**) Comparing with a similar spectrum; (**f**) Modified trend for the AI-assisted control.

**Figure 7 sensors-19-03440-f007:**
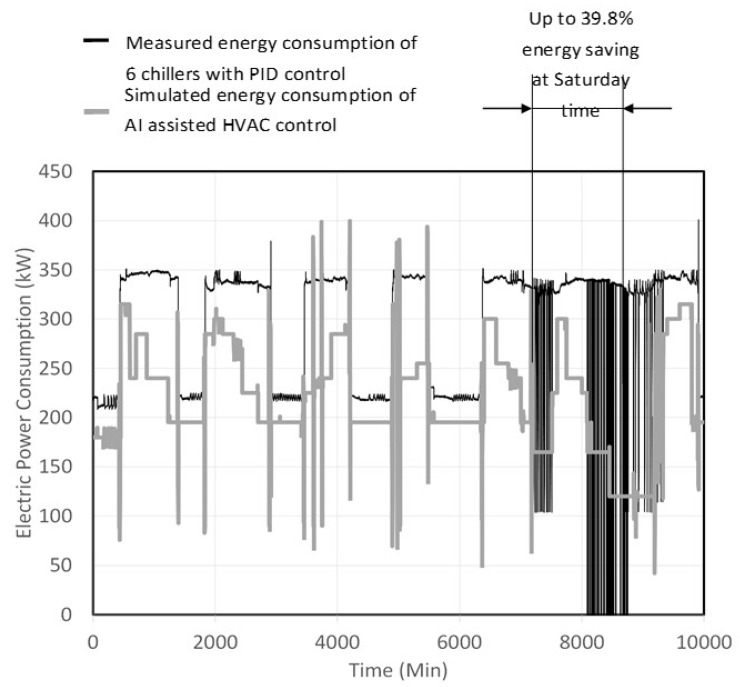
The energy savings of the AI-assisted control on the hottest week of 2017.

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
