# Peer review of "Artificial Intelligence Assisted Heating Ventilation and Air Conditioning Control and the Unmet Demand for Sensors: Part 2. Prior Information Notice (PIN) Sensor Design and Simulation Results"

_sensors, 2019, doi:10.3390/s19153440_

Round 1

Reviewer 1 Report

The paper deals with a subject that falls within the scope of the journal.

In the introduction (line 28) a reference is needed for Part 1.

The paragraph explaining Part 3 should be removed as or revised as future work.

The references in lines 44 to 48 should be removed as they were cited in Part 1. In line 48 the word cooling load is probably missing.

In line 210 the authors state that they use a COP of 3.5. A reference should be used (either from the technical characteristics of the actual system or from the literature, the following could be of help:

DOI: 10.1016/j.rser.2018.03.060

DOI: 10.1016/j.rser.2015.10.157)

Also, usually the COP changes depending on the outside temperature and humidity. This should also be included in the simulation as it impacts the electricity consumption.

Author Response

Dear editor:

Enclosed for your consideration is the answers for the reviewers’ comments of our manuscript, entitled " Artificial intelligence assisted heating ventilation and air conditioning control and the unmet demand for sensors: Part 2. prior information notice (PIN) sensor design and simulation results". The authors of this manuscript are Chin-Chi Cheng and Dasheng Lee. The revised version according to the reviewers’ comments is attached. Several English corrections are modified also.

Thank you very much for considering this article for your Journal.

Sincerely

Dasheng Lee

Distinguished Professor

Dept. of Energy and Refrigerating Air Conditioning Engineering

National  Taipei University of Technology

Phone: +886227712171 EXT 3510

Fax: +886227314919

Reviewer 1:

1. The paper deals with a subject that falls within the scope of the journal.

Answer: Thanks the reviewer’s recognition.

2. In the introduction (line 28) a reference is needed for Part 1.

Answer: Following the reviewer’s comment, a reference for Part 1 in line 28 of introduction is added.   

3. The paragraph explaining Part 3 should be removed as or revised as future work.

Answer: Following the reviewer’s comment, the paragraph explaining Part 3 has been revised as future work in the conclusions. The first paragraph in the introduction has been revised as following:

1. Introduction

This study continues the hypothesis proposed in Part 1 [1] that AI assisted HVAC control is a method of using predication to control and improve overall performance. However, due to the degree of prediction uncertainty being higher than that of a sensor, the capability to control system performance through this method has been limited. If there was a way to effectively improve prediction accuracies, the overall HVAC control performance could be improved. As far as the effect on energy savings is concerned, AI assisted control reached an energy savings of 57% and 44.64% compared to On-Off and PID control, respectively. Based on the above theoretically estimated results, an experimental case was selected to install an innovative sensor and calculate energy savings through simulation. The results were then compared with the theoretical estimation results. These are the main targets of this study, which discusses the sensor design and simulation results.

4. The references in lines 44 to 48 should be removed as they were cited in Part 1. In line 48 the word cooling load is probably missing.

Answer: Following the reviewer’s comment, the references in lines 44 to 48 have been removed, and the word cooling load in line 48 has been added. The second paragraph of introduction has been modified as following:

1. Introduction

Firstly, a literary review related to prediction uncertainty of controlling air-conditioners was done, including energy consumption; heating or cooling load, human behavior, thermal comfort index [2] and weather forecast. Referring to the literatures above and from part 1, the prediction uncertainty for the energy consumption, heating or cooling load is 7.46%,the prediction uncertainty for human behavior and thermal comfort is 14.50%,weather forecast uncertainty is 3.50%. The prediction uncertainties of temperature and humidity sensors, which are widely used in HVAC systems, are between 2.77 to 4.0 %. Besides weather forecast results, the prediction uncertainty of all the other criteria were higher than those of typical HVAC control sensors, clearly showing that the data gathered from the sensor is more accurate than predicted data.

5. In line 210 the authors state that they use a COP of 3.5. A reference should be used (either from the technical characteristics of the actual system or from the literature, the following could be of help:

DOI: 10.1016/j.rser.2018.03.060 

DOI: 10.1016/j.rser.2015.10.157)

Also, usually the COP changes depending on the outside temperature and humidity. This should also be included in the simulation as it impacts the electricity consumption.

Answer: Following the reviewer’s comment, in line 210 the references for the coefficient 3.5 which converts the unit RT to kW value have been added. During the simulation, the COP also varies with the outside temperature and humidity. The sixth paragraph of section 4 has been modified as following:

4. Simulation model and applied tools

The first part of equation (5) describes the real-time thermal mass change caused by pedestrian movement, the second and third parts describes the heat transmitting through the thermal resistance (wall) and air movement, respectively, from the outside temperature and humidity. Based on equation (5), temperature T1~T16 and relative humidity RH1~RH16 predicted by AI are able to calculate the heat flux needed for air conditioning system. That enables a model based predictive control and the energy consumption Epred can be calculated by

(6)

where COP with respect to every chiller is directly measured on line and the coefficient 3.5 is used to convert the unit RT to kW value [6][7]. During the simulation, the COP also varies with the outside temperature and humidity.

Reviewer 2 Report

Paper refers to PART 1 and PART 3 ... it is rare that you mention unpublished material. It is difficult to see the complete picture without seem all the material.

The paper is not so clear and should be partially rewritten.

Some examples :

“206 The control model shown in Figure 3 ….”

I cannot see any control model …. In figure 3

You should clarify equation (5)

“219  The basis of the PIN sensor design is that the daily collected sensor output data can be efficiently converted to trend and frequency.”

You should clearly specify what do you mean by trend an frequency

Why don't you implement digitally all the operations ? ....

Author Response

Dear editor:

Enclosed for your consideration is the answers for the reviewers’ comments of our manuscript, entitled " Artificial intelligence assisted heating ventilation and air conditioning control and the unmet demand for sensors: Part 2. prior information notice (PIN) sensor design and simulation results". The authors of this manuscript are Chin-Chi Cheng and Dasheng Lee. The revised version according to the reviewers’ comments is attached. Several English corrections are modified also.

Thank you very much for considering this article for your Journal.

Sincerely

Dasheng Lee

Distinguished Professor

Dept. of Energy and Refrigerating Air Conditioning Engineering

National  Taipei University of Technology

Phone: +886227712171 EXT 3510

Fax: +886227314919

Reviewer 2:

1. Paper refers to PART 1 and PART 3 ... it is rare that you mention unpublished material. It is difficult to see the complete picture without seem all the material.

Answer: Following the reviewer’s comment, the paragraph explaining Part 3 has been revised as future work in the conclusions. The first paragraph in the introduction has been revised as following:

1. Introduction

This study continues the hypothesis proposed in Part 1 [1] that AI assisted HVAC control is a method of using predication to control and improve overall performance. However, due to the degree of prediction uncertainty being higher than that of a sensor, the capability to control system performance through this method has been limited. If there was a way to effectively improve prediction accuracies, the overall HVAC control performance could be improved. As far as the effect on energy savings is concerned, AI assisted control reached an energy savings of 57% and 44.64% compared to On-Off and PID control, respectively. Based on the above theoretically estimated results, an experimental case was selected to install an innovative sensor and calculate energy savings through simulation. The results were then compared with the theoretical estimation results. These are the main targets of this study, which discusses the sensor design and simulation results.

2. The paper is not so clear and should be partially rewritten.

Some examples :

“206 The control model shown in Figure 3 ….”

 I cannot see any control model …. In figure 3

 You should clarify equation (5)

 Answer: Following the reviewer’s comment, line 206 has been modified as “The test site simulation shown in Figure 3 can be expressed by the following formula:”. The first part of equation (5) describes the real-time thermal mass change caused by pedestrian movement, and the second and third parts describes the heat transmitting through the thermal resistance (wall) and air movement, respectively, from the outside temperature and humidity. The sixth paragraph of section 4 has been modified as following:  

4. Simulation model and applied tools

The first part of equation (5) describes the real-time thermal mass change caused by pedestrian movement, the second and third parts describes the heat transmitting through the thermal resistance (wall) and air movement, respectively, from the outside temperature and humidity. Based on equation (5), temperature T1~T16 and relative humidity RH1~RH16 predicted by AI are able to calculate the heat flux needed for air conditioning system. That enables a model based predictive control and the energy consumption Epred can be calculated by

(6)

where COP with respect to every chiller is directly measured on line and the coefficient 3.5 is used to convert the unit RT to kW value [6][7]. During the simulation, the COP also varies with the outside temperature and humidity.

3. “219  The basis of the PIN sensor design is that the daily collected sensor output data can be efficiently converted to trend and frequency.”

You should clearly specify what do you mean by trend and frequency

Why don't you implement digitally all the operations ? ....

 Answer: Following the reviewer’s comment, the sensor output signal contains the DC and AC information. The DC information will be acquired by the trend filter, and the AC information will be filtered out by the frequency acquiring circuit. This explanation has been presented in the fourth paragraph of section 3. The digitized signal output is not able to trace these two facts. The second paragraph of section 5.1 has been modified as following:

5.1. Big data collection

Between 2012 and 2017, a total of 630,072,000 data was collected to design the PIN sensor and verify AI assisted HVAC control performance. Figure 4 (a) shows the temperature sensor data at 1F lobby of the test site location in one day of 2012.

The basis of the PIN sensor design is that the daily collected sensor output data can be efficiently converted to trend and frequency, which are the DC and AC components of sensor output data. According to the variation of trend, the collected sensor output data can be divided into 6 segments, as shown in Figure 4 (b)-(h). After further analysis of each segment, the trend and frequency response of sensor output data have close relationship, which is the design basis of the PIN sensor. Matlab Simulink is utilized to design the analog and digital circuitry, and verify whether or not the designed circuit can convert the sensor output signal to segment length time, trend and frequency.

Round 2

Reviewer 1 Report

The authors have dealt with the comments raised.

Author Response

Reviewer 1:

1.      The authors have dealt with the comments raised.

Answer: Thanks the reviewer’s recognition.

Reviewer 2 Report

Although the introduction has been modified the paper is basically the same.

It still do not understand the main goal of the paper.

How the Figure 2 circuit relates to the other develops.

The paper should be better explained before publishing
